# Study International Multicentric Pancreatic Left Resections (SIMPLR): Does Surgical Approach Matter?

**DOI:** 10.3390/cancers16051051

**Published:** 2024-03-05

**Authors:** Sara Acciuffi, Mohammed Abu Hilal, Clarissa Ferrari, Sara Al-Madhi, Marc-Anthony Chouillard, Nouredin Messaoudi, Roland S. Croner, Andrew A. Gumbs

**Affiliations:** 1Department of General-, Visceral-, Vascular- and Transplantation Surgery, University of Magdeburg, Leipziger Str. 44, 39120 Magdeburg, Germany; sara.acciuffi@med.ovgu.de (S.A.); sara.al-madhi@med.ovgu.de (S.A.-M.); roland.croner@med.ovgu.de (R.S.C.); 2Hepatobiliopancreatic, Robotic and Minimally Invasive Surgery Unit, Fondazione Poliambulanza Istituto Ospedaliero, Via Bissolati 57, 25124 Brescia, Italy; abuhilal9@gmail.com; 3Research and Clinical Trials Office, Fondazione Poliambulanza Istituto Ospedaliero, Via Bissolati 57, 25124 Brescia, Italy; clarissa.ferrari@poliambulanza.it; 4Hepatobiliopancreatic Surgery, Université de Paris Cité, 85 boulevard Saint-Germain, 75006 Paris, France; marcanthony.chouillard@pm.me; 5Department of Hepatopancreatobiliary Surgery, Vrije Universiteit Brussel, Universitair Ziekenhuis Brussel and Europe Hospitals, Laarbeeklaan 101, 1090 Brussels, Belgium; nouredin.messaoudi@uzbrussel.be; 6Department of Advanced & Minimally Invasive Surgery, American Hospital of Tbilisi, 17 Ushangi Chkheidze Street, Tbilisi 0102, Georgia

**Keywords:** left pancreatic resection, laparoscopy, robotic surgery, minimally invasive, multicentric, international

## Abstract

**Simple Summary:**

Nowadays: minimal invasive distal pancreatectomy is becoming the standard approach for this procedure. Nevertheless, empirical evidence is still needed to validate the advantages associated with the various surgical approaches. This international retrospective multicenter cohort study conducted at three high-volume centers for HPB surgery attempted to compare the perioperative and oncological outcomes of the three primary surgical techniques—open, laparoscopic, and robotic—using propensity score matching analysis. The laparoscopic approach demonstrated notable benefits, including shorter operative times, lower blood loss, and reduced duration of both ICU and hospital stays. Furthermore, the robotic approach exhibited a significantly lower incidence of POPF. Notably, all three techniques demonstrated comparable levels of oncological safety, morbidity, and mortality.

**Abstract:**

Background: Minimally invasive surgery is increasingly preferred for left-sided pancreatic resections. The SIMPLR study aims to compare open, laparoscopic, and robotic approaches using propensity score matching analysis. Methods: This study included 258 patients with tumors of the left side of the pancreas who underwent surgery between 2016 and 2020 at three high-volume centers. The patients were divided into three groups based on their surgical approach and matched in a 1:1 ratio. Results: The open group had significantly higher estimated blood loss (620 mL vs. 320 mL, *p* < 0.001), longer operative time (273 vs. 216 min, *p* = 0.003), and longer hospital stays (16.9 vs. 6.81 days, *p* < 0.001) compared to the laparoscopic group. There was no difference in lymph node yield or resection status. When comparing open and robotic groups, the robotic procedures yielded a higher number of lymph nodes (24.9 vs. 15.2, *p* = 0.011) without being significantly longer. The laparoscopic group had a shorter operative time (210 vs. 340 min, *p* < 0.001), shorter ICU stays (0.63 vs. 1.64 days, *p* < 0.001), and shorter hospital stays (6.61 vs. 11.8 days, *p* < 0.001) when compared to the robotic group. There was no difference in morbidity or mortality between the three techniques. Conclusion: The laparoscopic approach exhibits short-term benefits. The three techniques are equivalent in terms of oncological safety, morbidity, and mortality.

## 1. Introduction

Since the first documented laparoscopic distal pancreatectomy in 1993 [1] followed by the pioneering report of a robotic distal pancreatectomy in 2004 [2], minimally invasive approaches have garnered heightened interest and have become the gold standard surgical technique for this procedure [3,4]. A recent international survey reported that 94.6% of surgeons consider minimally invasive surgery (MIS) superior to the open approach when performing distal pancreatectomy (DP) [5]. Notably, laparoscopy remains the preferred technique due to its lower cost and faster set-up time [5]. Three randomized controlled trials have demonstrated the short-term clinical advantages of MIS, including reduced length of hospital stay (LOS) and decreased blood loss [6,7,8]. The minimally invasive approach for DP has proven to be safe and viable with comparable morbidity, mortality, and oncologic outcomes, even for cases of pancreatic cancer [8].

Until a few years ago, in the statistical comparison to open surgery, laparoscopic and robotic approaches were considered as part of a single entity, that of MIS. Given the significant differences between the two techniques in terms of costs, set-up time, and operative interface, recently the interest has shifted towards the direct comparison between laparoscopy and robotic surgery [9,10,11,12,13]. Since the first meta-analysis of 2016 investigating the difference between laparoscopic and robotic approaches for DP, with 7 studies including 568 patients [13], an increasing number of studies have been published to further investigate these 2 modalities [12,14,15,16,17,18,19,20,21,22]. The recent review of Van Ramshorst et al. enrolled 43 studies and a total of 6757 patients showing that the use of a minimal invasive approach for this type of operation has experienced a monstrous growth of interest in the last decade [12].

Two recent multicenter cohort studies have concurred that robotic surgery for DP is associated with lower conversion rates, higher lymph node yield, similar R0 resection rates, and longer operative times compared to laparoscopy [9,11]. Nevertheless, results are not uniform in terms of postoperative morbidity and LOS. Robotic surgery is increasingly becoming the preferred technique due to greater instrument dexterity, 3D high-definition visualization, and tremor filtration, theoretically enabling more surgeons to tackle more technically challenging procedures [5]. On the other side of the argument, robotic approaches mean a significantly higher cost for healthcare systems [23,24]. Therefore, further studies are needed to substantiate the benefits of these individual strategies. The aim of this study is to compare the three most commonly used surgical techniques for DP (open, laparoscopy, and robotic) investigating the perioperative and oncological outcomes using a propensity match score analysis.

## 2. Materials and Methods

The following is an international retrospective multicenter cohort study conducted at three high-volume centers for hepatopancreatobiliary (HPB) surgery, located in Italy, France, and Germany. The study was conducted in accordance with the Strengthening the Reporting of Observational Studies in Epidemiology (STROBE) guidelines [25].

### 2.1. Patient Recruitment

All consecutive patients aged 18 years or older, who underwent an open, laparoscopic, or robotic DP between January 2016 and December 2020 for a benign or malignant neoplasm of the body or tail of the pancreas were included in the study. Patients were categorized into three groups: those who underwent an open DP (ODP), those who underwent a laparoscopic DP (LDP), and those who underwent a robotic DP (RDP). All pancreatic resections were performed at high-volume centers by experienced HPB surgeons with proficiency in both open and minimally invasive surgery, having completed at least 50 laparoscopic and/or robotic DP procedures. Each center was provided with a database to record all necessary perioperative variables. The completed databases were summarized by the study coordinator. Patients were followed up for a minimum of 3 months post-surgery.

### 2.2. Surgical Technique

We evaluated three distinct surgical approaches: open, laparoscopic, and robotic. The surgical procedures were standardized across the centers. In all procedures, surgeons adopted a “medial to lateral” approach, implementing parenchymal-preserving techniques for benign neoplasias and opting for left pancreatectomy with splenectomy in cases of cancer. For malignant lesions, the portal vein served as the standard reference plane for resection. Energy devices were not standardized across centers and were based on individual surgeon preferences. Pancreatic transections were uniformly conducted using a linear gastrointestinal anastomotic stapler, whether automatic or manual. The specimen extraction in case of a minimally invasive procedure was accomplished through either a Pfannenstiel-Kerr incision or by enlarging one of the trocar sites. The laparoscopic operations were fully performed with laparoscopic ports. The robotic approach was “pure robotic” as defined in the European guidelines: the system (Intuitive Surgical, Inc., Sunnyvale, CA, USA) is docked at the beginning of the operation, the procedure is performed through 3–4 robotic ports and 1 or more laparoscopic ports and no laparoscopic energy device was used [26]. All the centers used the da Vinci System (Intuitive Surgical, Inc., Sunnyvale, CA, USA) for robotic surgery.

### 2.3. Definitions

High-volume centers are considered centers with at least an average of 20 DP performed per year [26]. Converted procedures were defined as formal not-intended conversions to laparotomy at any stage of the procedures for any reason (bleeding, vascular resection, difficult anastomosis, not-progression, etc.) [26]. In the pathological analysis, R0 resection is defined as microscopic radical resection of at least 1 mm between the tumor at the transection or retroperitoneal margin. Postoperative complications were classified using the Clavien–Dindo Classification [27] and severe complications had a Clavien–Dindo score of 3a or above. Specific pancreatic complications, such as postoperative pancreatic fistula (POPF), were classified following the most recent definitions of the International Study Group on Pancreatic Surgery [28]. Morbidity and mortality were recorded for up to 90 days after surgery.

### 2.4. Endpoints

The primary endpoints were perioperative parameters such as intraoperative blood loss, operative time, length of hospital stay (LOS), and oncologic results (R0 resection and lymph nodes yield). Secondary endpoints included severe postoperative complications, POPF, and 30-day and 90-day mortality rates.

### 2.5. Statistical Analysis

Categorical data (nominal/ordinal) are presented as absolute (n) and percentage values (%). Differences between the groups were tested using Pearson’s Chi-squared test (or Fisher’s exact test when appropriate). Continuous variable distributions were described by mean and standard deviation (SD). Differences between continuous variables were analyzed using a *t*-test or corresponding non-parametric Mann–Whitney U-test for comparing two groups of Gaussian or non-Gaussian distributed variables, respectively. Differences among all groups were evaluated by an ANOVA or a Kruskal–Wallis test. Post-hoc multiple comparisons were adjusted by the Bonferroni correction. The use of parametric or corresponding non-parametric tests was reported in the tables of results. The evaluation of means and SDs allowed us to assess the potential skewness of the variable and to justify the appropriateness of the applied non-parametric test. The propensity score (PS) was calculated using a logistic regression model. Potential confounding variables/predictors were chosen in agreement with all HPB surgeons of the participating study centers. Those predictors comprise age, gender, ASA (American Society of Anesthesiology) score, prevalence of previous abdominal surgery, prevalence of previous chemotherapy, type of lesion (benign, malignant), number of resected tumors, maximum size of resected tumors and BMI (body mass index). The surgical approach (open vs. laparoscopic vs. robotic) was used as the dependent group variable. We set the matching tolerance to 0.2. Matching was performed in a 1:1 ratio of nearest neighbor without replacements; and several PS settings (distances, models, link functions) were carried out in order to find the best matching. The patients of each surgical approach were matched in turn to each other groups. We excluded patients with incomplete information on predictors as needed for the appropriate application of the propensity score matching procedure; in addition, we confirmed that the incomplete data was not different from the rest of the samples in terms of predictor features. In case of failed matches for some covariates, the comparison of the outcomes between groups was adjusted for those covariates by using generalized linear models. Data analysis was performed using the R software v.4.2.2 (https://cran.r-project.org/). For the propensity score analysis, the specific R-package ‘Matchlt’ was used (https://cran.r-project.org/web/packages/MatchIt/index.html).

## 3. Results

### 3.1. Participant and Descriptive Data

From January 2016 to December 2020, a total of 258 patients were enrolled in the study. The ODP group included 34 patients, the LDP group included 192 patients, and the RDP group included 32 patients. Out of the total 258 patients included in the study, 153 (59.3%) were female, while 105 (40.7%) were male. The average age of the patients was 62 years, with a mean body mass index (BMI) of 27.9 kg/m^2^. Among the cohort, 32 patients (12.4%) had an ASA score of 1, 157 patients (60.8%) had an ASA score of 2, 67 patients (26%) had an ASA score of 3, and 2 patients (0.8%) had an ASA score of 4. Of the 258 patients, 193 (74.8%) had not undergone prior abdominal surgery. Neoadjuvant chemotherapy was administered to only five patients (1.9%). Of the 258 tumors, 140 (55%) were identified as malignant lesions, with an average diameter of 39.6 mm. Prior to matching, there were no demographic differences observed in terms of sex, age, and type of lesion (Table 1). However, patients in the ODP group exhibited significantly larger tumor lesions, higher BMI and ASA scores, as well as a significantly higher percentage of patients who had received chemotherapy before resection. The LDP group had a significantly lower percentage of patients with a history of previous abdominal surgery.

For the application of the propensity score (PS) matching, complete data are needed so we excluded patients with incomplete information on predictors. Thus, the sample used for the PS was composed of 31 ODP cases, 162 LDP cases, and 28 RDP cases. The incomplete data showed no differences in terms of predictors used in the matching procedure so no bias was introduced by data exclusion. The size of the surgical approach groups, although slightly reduced, was enough to ensure sufficient power also when there was a quite low effect size.

### 3.2. Comparison between Open (ODP) and Laparoscopic (LDP) Groups

Table 2 presents the confounding and outcome variables of patients who underwent open and laparoscopic interventions before and after propensity score matching. A total of 31 patients were matched in a 1:1 ratio. After matching, there were no differences in demographic variables. Additionally, there were no discrepancies in oncologic variables; the two techniques exhibited similar lymph node (15.6 vs. 13, *p* = 0.804) and pathologic lymph node yields (1.34 vs. 1.14, *p* = 0.479) as well as similar resection status (R0/R1 25/5 vs. 24/7, *p* = 0.796). However, estimated blood loss (620 vs. 320 mL, *p* < 0.001), operative time (273 vs. 216 min, *p* = 0.003), hospital stay (16.9 vs. 6.81 days, *p* < 0.001), and intensive care unit (ICU) stay (2.63 vs. 0.13 days, *p* < 0.001) were significantly lower in patients of the LDP group. There was no significant difference in POPF rate between the two methods (*p* = 0.279). There were no significant differences in major morbidity (*p* = 0.983) or 30-day (*p* = 0.996) or 90-day mortality rates (*p* = 0.985).

### 3.3. Comparison between Open (ODP) and Robotic Group (RDP)

Table 3 presents the confounding and outcome variables of patients who underwent open and robotic interventions before and after propensity score matching. A total of 28 patients were successfully matched. After matching, there were no statistically significant differences in sex, age, BMI, ASA score, history of prior abdominal surgery, percentage of patients receiving pre-operative chemotherapy, or type of lesion. The ODP group removed significantly larger tumors (49.7 vs. 31.9 mm, *p* = 0.003). Regarding surgical outcomes, there were no statistically significant differences in resection margins (*p* = 0.713), but during robotic operations, a higher number of lymph nodes were removed compared to ODP (24.9 vs. 15.2, *p* = 0.011). Pathologic lymph node yields didn’t differ between the two groups (*p* = 0.763). The ODP tended to be faster than the RDP (265 vs. 340 min), although this difference did not reach statistical significance (*p* = 0.091). Patients in the RDP group experienced a significantly shorter LOS compared to those in the ODP group (11.8 vs. 17.5 days, *p* = 0.001). Patients in the ODP group had a higher incidence of Grade B/C POPF (6 vs. 0, *p* = 0.057), even if the discrepancy didn’t reach statistical significance. There were no significant discrepancies in estimated blood loss (603 vs. 262 mL, *p* = 0.126), ICU stay (2.70 vs. 1.64 days, *p* = 0.144), major morbidity (*p* = 0.384), or 30-day (*p* = 0.997) and 90-day mortality rates (*p* = 0.51).

### 3.4. Comparison between Laparoscopic (LDP) and Robotic (RDP) Groups

Table 4 presents the confounding and outcome variables of patients who underwent laparoscopic and robotic procedures before and after propensity score matching. We matched 28 patients. After matching, the demographic variables did not differ between the two groups. In the RDP group, a significantly higher number of lymph nodes were removed (*p* < 0.001), even though the number of pathological lymph nodes did not differ (0.55 vs. 1.8, *p* = 0.206). There was no significant difference in resection status (R0/R1: 27/1 vs. 26/2, *p* = 0.993). The LDP group had a significantly shorter operative time (210 vs. 340 min, *p* < 0.001), postoperative ICU stay (0.63 vs. 1.64 days, *p* < 0.001), and LOS (6.61 vs. 11.8 days, *p* < 0.001). The robotic group had significantly fewer POPF (*p* < 0.001) and fewer clinically relevant POPF grade B/C than the LDP group (*p* = 0.014). There were no differences in estimated blood loss (*p* = 0.817), major morbidity (*p* = 0.561), 30-day mortality (*p* = 1), and 90-day mortality (*p* = 1). Moreover, there were no statistically significant differences in conversion rate between the two approaches (*p* = 0.156).

## 4. Discussion

The international guidelines of Miami in 2020 and the Brescia guidelines in 2023 already confirmed that minimally invasive distal pancreatectomy is superior to open approaches and offers significant short-term postoperative benefits in cases of benign and low malignant lesions [4,26]. The LEOPARD randomized control trial demonstrated that MIS has reduced time to functional recovery, lower operative blood loss, shorter LOS, and a better postoperative quality of life than open surgery, all without an increase in costs [7]. Additionally, the more recent DIPLOMA trial found no discrepancy in postoperative morbidity and survival between the two techniques even for patients with pancreatic cancer [8].

Nowadays, with the term “minimally invasive surgery” we are encompassing a diverse range of approaches, each with its own distinct merits and drawbacks. Therefore, it is pertinent to assess each of these approaches separately to clarify their distinct benefits and potential drawbacks. In the present study, we performed an international multicenter comparison between laparoscopic, robotic, and open approaches for DP. To ensure a more uniform and comparable patient cohort, we employed propensity score matching. Our findings indicate that the laparoscopic technique demonstrates superior short-term advantages, including reduced hospital and ICU stays, as well as shorter operative time when compared to both other techniques. Importantly, this didn´t affect the oncological efficacy since the resection status and pathological lymph node yields didn´t differ. Notably, the robotic technique exhibited a significantly lower incidence of POPF compared to laparoscopy, which is crucial since postoperative morbidity is indeed associated with adjuvant treatment delay or omission [29,30]. Despite these facts, there were no observable differences in terms of major morbidity and mortality among the three procedures.

Previous cohort studies support our observation that laparoscopic surgery generally entails shorter operative time compared to robotic surgery [9,11,24,31,32]. One potential factor contributing to this observation is the lack of a clear distinction between effective operative time and docking time in the robotic approach. In robotic surgery, the duration of preparation and docking time are notably longer than in laparoscopy or open approaches, even when performed by experienced surgical teams; however, the difference is quite big to be justified only by docking time. Furthermore, our study involved surgeons experienced in pancreatic laparoscopic surgery, accounting for the shorter operative time compared to robotic surgery. Given the recent introduction of robotic surgery at the enrolled centers, surgeons may still be in the learning curve phase, contributing to the observed differences.

As highlighted in previous studies [6,7,33], patients undergoing laparoscopic procedures in our cohort demonstrated statistically significant reduced blood loss when compared to those undergoing open surgery. This presents a noteworthy benefit of MIS, given that excessive blood loss is an established risk factor for pancreatic fistula [34]. However, as emphasized by Perri et al., a standardization of the intraoperative blood loss estimation is urgently needed to enhance the reliability of results [35].

Compared to the robotic approach, laparoscopic surgery had significantly shorter LOS, this indicates that laparoscopic surgeons with experience and skills may be as good as the robot or maybe even better. Even if not significant, the RDP group experienced higher postoperative complications, which could certainly contribute to a prolonged hospital stay. Additionally, different discharge policies of the different centers could also have influenced this variable. Previous studies by Souche et al. found no difference in LOS [32], and Liu et al. demonstrated a shorter LOS for robotic surgery [36]. A recent study by Chen et al. had shown similar results to our study. Two recent meta-analyses found no difference in LOS between laparoscopic and robotic groups [12,19]. The difference in hospitalization duration is especially noticeable when comparing laparoscopic or robotic approaches to open procedures. Patients undergoing open DP experience a hospital stay more than twice as long as those undergoing laparoscopic procedures. This variation has been demonstrated in previous studies [6,7,33]. The LEOPARD trial showed a reduction of 33% in hospitalization duration with minimally invasive surgery [7]. This difference in LOS related to open surgery may be attributed to factors such as pain management and delayed functional recovery, which play an important role in the economic aspects of healthcare. Both Abu Hilal et al. and Limongelli et al. demonstrated a significant reduction in postoperative costs with laparoscopic DP in comparison to open surgery [37,38].

In our study, patients who underwent robotic-assisted surgery exhibited a significantly lower incidence of clinically relevant POPF compared to those who underwent laparoscopic procedures. Other previous cohort studies and a meta-analysis have failed to identify a significant difference in the occurrence of POPF Grade B/C between laparoscopic and robotic techniques [9,12,39]. However, within our study group, this discrepancy did not result in a significant impact on postoperative comorbidities or mortality.

Moreover, the robotic approach offers enhanced visualization in a magnified three-dimensional field, facilitating meticulous precision in isolating vessels. This precision is particularly advantageous for managing intraoperative bleeding, likely contributing to the lower reported conversion rates mentioned in the literature [9,12,23,36,40,41]. The large meta-analysis of Ramshorst et al. that compared 2514 patients who underwent RDP and 4243 patients who underwent LDP, found lower conversion rates in their robotic group with a higher rate of spleen preservation either with or without splenic vessel preservation [12]. In contrast to the previously published literature, in our cohort after matching the groups, the two techniques did not show a significant difference in conversion rate. This may depend on the fact that the surgeons have extensive experience in laparoscopy which may mitigate the lack of robotic facilities. However, we were not able to differentiate “urgent conversion” which is caused by an emergency, from “non-urgent conversion”, which is related to unexpected factors like tumor size, adhesions, etc. Considering that only urgent conversions have a negative impact on patient outcomes, future studies could improve by reporting these two situations separately, as suggested in the latest guidelines [26].

Notably, both minimally invasive approaches did not differ from the open approach in terms of oncological efficacy. We reached an overall R0 resection of 87%, 93%, and 83% for the LDP, RDP, and ODP groups, respectively. Despite the multicenter nature of the study and the risk of heterogeneity in the pathological analysis of the specimen, no differences in R0 resection were observed after matching the groups. Additionally, although the robotic technique allows more lymph node yield, there were no significant differences in the rate of pathological lymph nodes among the three groups. The DIPLOMA randomized control trial confirmed our results and demonstrated a non-inferiority of MIS for patients with pancreatic cancer in terms of radical resection, lymph node yield, overall survival, and disease-free survival [8]. Of note, when we analyzed the oncological data, we only considered patients with cancer. As concluded in previous studies [6,7,8,9,11,33,39], the 3 surgical approaches did not differ in terms of postoperative major morbidity and 30-day and 90-day mortality, making every approach safe and feasible for all indications.

Nowadays, minimally invasive distal pancreatectomy is becoming the standard approach for this type of operation, because it offers notable short-term advantages over the open approach, including a shorter LOS, reduced ICU stay, shorter operative time, and minimized blood loss. These benefits are achieved with equivalent levels of safety, morbidity, and mortality, which significantly influence patient outcomes and postoperative quality of life. In choosing between the laparoscopic or robotic approach, the proficiency of the surgeons should play a fundamental role.

As highlighted in the recent European guidelines on minimally invasive pancreatic surgery, the center volume and the experience of the surgical team are fundamental factors relevant to patient outcomes [26]. Hence, the choice of the most appropriate approach is based on the surgeon’s decision, considering not only patient and tumor characteristics but also the proficiency of the surgical team.

The primary limitation of the study lies in its retrospective nature. The application of propensity score matching analysis helped alleviate biases related to group allocation and the selection of surgical approach. Another limitation is the international and multicenter design, which introduces variations in hospital policies and healthcare systems that may affect the quality of the results. Crucial data, including details about distinctions between urgent and non-urgent conversions, are lacking. These factors should be considered when interpreting the findings of this study.

Finally, the integration of artificial intelligence (AI) in pancreatic cancer surgery is set to transform decision-making in the coming years by tailoring open, laparoscopic, and robotic strategies based on detailed analyses of clinical, pathological, and radiological data [42,43,44]. AI algorithms, acting as robust decision support tools, address the complex dilemma of selecting the most suitable surgical approach by enhancing preoperative planning precision in left pancreatic tumors [42]. Concurrently, initiatives like the AiRGOS project exemplify the evolving landscape of personalized medicine [42]. This project aims to develop an AI algorithm that combines radiologic imaging, whole genomic sequencing, oncopathological data, and therapeutic responses to provide a precision therapeutic plan based on the patient’s entire oncologic profile [42]. As AI technology progresses, these advancements hold the promise of refining and optimizing surgical approaches for individual patients undergoing pancreatic cancer surgery [42,43]; however, many ethical and technical obstacles still remain [44,45,46,47].

## 5. Conclusions

The findings from this study indicate that the three investigated surgical techniques, when performed in high-volume centers, are equivalent in terms of oncological safety and postoperative morbidity and mortality. However, the laparoscopic approach continues to exhibit short-term benefits, including shorter operative times, reduced blood loss as well as reduced hospital and ICU stays, contributing to faster functional recovery. These factors are crucial considerations for the overall quality of life of the patients.

While robotic surgery is increasingly becoming the preferred approach for such procedures, enabling the handling of more complex cases, further studies are needed to understand the tangible benefits offered by this technique. Especially since skilled and experienced laparoscopic surgeons may be able to have equivalent or better outcomes when compared to the robotic approach.

## Figures and Tables

**Table 1 cancers-16-01051-t001:** Demographic data by surgical approach.

	ODP (*n* = 34)	LDP (*n* = 192)	RDP (*n* = 32)	*p*-Value	Overall (*n* = 258)
Gender					
Female	16 (47.1%)	114 (59.4%)	23 (71.9%)	0.055	153 (59.3%)
Male	18 (52.9%)	77 (40.1%)	9 (28.1%)		105 (40.7%)
Age (years)					
Mean (SD)	62.9 (9.55)	62.3 (14.3)	61.3 (17.2)	0.145	62 (14.0)
BMI (kg/m^2^)					
Mean (SD)	30.3 (4.56)	27.8 (4.20)	25.7 (5.48)	<0.001	27.9 (4.84)
ASA Score					
1	0 (0%)	29 (15.1%)	2 (6.3%)	<0.001	32 (12.4%)
2	20 (58.8%)	120 (62.5%)	17 (53.1%)		157 (60.8%)
3	13 (38.2%)	41 (21.4%)	13 (40.6%)		67 (26%)
4	1 (2.9%)	1 (0.5%)	0 (0%)		2 (0.8%)
Prior Abdominal Surgery					
No	19 (55.9%)	153 (79.7%)	21 (65.6%)	<0.001	193 (74.8%)
Yes	15 (44.1%)	35 (18.2%)	11 (34.4%)		65 (25.2%)
Neoadjuvant Chemotherapy					
No	31 (91.2%)	183 (95.3%)	32 (100%)	0.004	253 (98.1%)
Yes	3 (8.8%)	2 (1.0%)	0 (0%)		5 (1.9%)
Type of Lesion					
Benign	13 (38.2%)	88 (45.8%)	15 (46.9%)	0.837	118 (45%)
Malignant	21 (61.8%)	103 (53.6%)	16 (50.0%)		140 (55%)
Diameter of resected tumor (mm)					
Mean (SD)	51.6 (33.1)	35.2 (20.3)	31.9 (25.7)	<0.001	39.6 (22.7)

Abbreviation. ODP: open distal pancreatectomy; LDP: laparoscopic distal pancreatectomy; RDP: robotic distal pancreatectomy.

**Table 2 cancers-16-01051-t002:** Open vs. Laparoscopic distal pancreatectomy.

	Before Matching			After Matching		
	ODP (*n* = 31)	LDP (*n* = 162)	*p* Value	ODP (*n* = 31)	LDP (*n* = 31)	*p* Value
Gender						
Female	14 (45.2%)	95 (58.6%)	0.234	14 (45.2%)	16 (51.6%)	0.799
Male	17 (54.8%)	67 (41.4%)		17 (54.8%)	15(48.4%)	
Age (years)						
Mean (SD)	63.3 (9.34)	61.7 (14.7)	0.417	63.3 (9.34)	62.5 (13.4)	0.578
BMI (kg/m^2^)						
Mean (SD)	30.3 (4.71)	28.1 (4.18)	0.020	30.3 (4.71)	30 (4.70)	0.894
ASA Score						
1	0 (0%)	23 (14.2%)	0.068	0 (0%)	0 (0%)	
2	19 (61.3%)	102 (63%)		19 (61.3%)	18 (58.1%)	0.978
3	12 (38.7%)	36 (22.2%)		12 (38.7%)	13 (41.9%)	
4	0 (0%)	1 (0.6%)		0 (0%)	0 (0%)	
Prior Abdominal Surgery						
No	18 (58.1%)	132 (81.5%)	0.008	18 (58.1%)	20 (64.5%)	0.794
Yes	13 (41.9%)	30 (18.5%)		13 (41.9%)	11 (35.5%)	
Neoadjuvant Chemotherapy						
No	28 (90.3%)	160 (98.8%)	0.034	28 (90.3%)	30 (96.8%)	0.605
Yes	3 (9.7%)	2 (1.2%)		3 (9.7%)	1 (3.2%)	
Type of lesion						
Benign	10 (32.3%)	72 (44.4%)	0.289	10 (32.3%)	5 (16.1%)	0.236
Malignant	21 (67.7%)	90 (55.6%)		21 (67.7%)	26 (83.9%)	
Diameter of resected tumor (mm)						
Mean (SD)	51.6 (33.1)	35.8 (19.9)	0.014	51.6 (33.1)	44.9 (23.6)	0.535
number LNN Retrieved						
Mean (SD)	15.6 (12.8)	10.6 (7.61)	0.062	15.6 (12.8)	13.0 (6.57)	0.804 ^MW^
number Pathologic.LNN						
Mean (SD)	1.34 (2.21)	0.94 (1.32)	0.364	1.34 (2.21)	1.14 (1.10)	0.479 ^MW^
Intraoperative blood transfusion						
No	25 (80.6%)	152 (93.8%)	0.361	25 (80.6%)	28 (90.3%)	0.936
Yes	3 (9.7%)	7 (4.3%)		3 (9.7%)	2 (6.5%)	
Postoperative ICU stay (days)						
Mean (SD)	2.63 (2.98)	0.26 (1.18)	<0.001	2.63 (2.98)	0.13 (0.72)	<0.001 ^MW^
POPF &			0.085			0.279
None	21 (67.7%)	86 (53.1%)	ns	21 (67.7%)	18 (58.1%)	Ns
Biochemical leak	4 (12.9%)	53 (32.7%)	ns	4 (12.9%)	9 (29%)	Ns
Grade B/C	6 (19.4%)	23 (14.2%)	ns	6 (19.4%)	4 (12.7%)	Ns
Resection margin						
R0	25 (83.3%)	141 (87%)	0.799	25 (83.3%)	24 (77.4%)	0.796
R1	5 (16.7%)	21 (13%)		5 (16.7%)	7 (22.6%)	
Operative time (minute)						
Mean (SD)	273 (80.1)	210 (73.1)	<0.001	273 (80.1)	216 (83.9)	0.003
Estimated blood loss (mL)						
Mean (SD)	620 (451)	282 (355)	<0.001	620 (451)	320 (344)	<0.001
Postoperative hospital stay (days)						
Mean (SD)	16.9 (12.4)	6.38 (5.01)	<0.001	16.9 (12.4)	6.81 (6.32)	<0.001 ^MW^
Postoperative Clavien-Dindo morbidity ≥ grade 3						
No	27 (87.1%)	139 (85.8%)	1	27 (87.1%)	26 (83.9%)	0.983
Yes	4 (12.9%)	23 (14.2%)		4 (12.9%)	5 (16.1%)	
30-day mortality						
No	29 (93.5%)	160 (58.8%)	0.960 #	29 (93.5%)	30 (96.8%)	0.996
Yes	1 (3.2%)	2 (1.2%)		1 (3.2%)	1 (3.2%)	
90-day mortality						
No	16 (51.6%)	160 (98.2%)	0.985 #	16 (51.6%)	30 (96.8%)	0.985 #
Yes	0 (0%)	2 (1.2%)		0 (0%)	1 (3.2%)	

Abbreviation. ODP: open distal pancreatectomy; LDP: laparoscopic distal pancreatectomy; RDP: robotic distal pancreatectomy. ICU = intensive care unit, LNN = lymph node, SD = standard deviation; POPF: postoperative pancreatic fistula. ns: not significant at level 0.05. MW: Mann–Whitney test *p*-value; #: chi-squared test on available (excluding missing) data; &: general chi-square *p*-value is reported in the line of the name of the variable. Post-hoc chi-square *p*-values, adjusted for Bonferroni correction, are reported in the lines corresponding to each category of the variable.

**Table 3 cancers-16-01051-t003:** Open vs. Robotic distal pancreatectomy.

	Before Matching			After Matching		
	ODP (*n* = 31)	RDP (*n* = 28)	*p* Value	ODP (*n* = 28)	RDP (*n* = 28)	*p* Value
Gender						
Female	14 (45.2%)	21 (75%)	0.038	14 (50%)	21 (75.0%)	0.098
Male	17 (54.8%)	7 (25.0%)		14 (50%)	7 (25.0%)	
Age (years)						
Mean (SD)	63.3 (9.34)	60.8 (18.1)	0.504	62.9 (9.62)	60.8 (18.1)	0.857
BMI (kg/m^2^)						
Mean (SD)	30.3 (4.71)	25.7 (5.41)	0.001	29.8 (4.72)	26.7 (5.65)	0.072
ASA Score						
1	0 (0%)	2 (7.1%)	0.299	0 (0%)	2 (7.1%)	0.311
2	19 (61.3%)	14 (50.0%)		17 (60.7%)	14 (50.0%)	
3	12 (38.7%)	12 (42.9%)		11 (39.3%)	12 (42.9%)	
4	0 (0%)	0 (0%)		0 (0%)	0 (0%)	
Prior Abdominal Surgery						
No	18 (58.1%)	19 (67.9%)	0.612	18 (64.3%)	19 (67.9%)	0.987
Yes	13 (41.9%)	9 (32.1%)		10 (35.7%)	9 (32.1%)	
Neoadjuvant Chemotherapy						
No	28 (90.3%)	28 (100%)	0.241	28 (100%)	28 (100%)	1
Yes	3 (9.7%)	0 (0%)		0 (0%)	0 (0%)	
Type of lesion						
Benign	10 (32.3%)	14 (50.0%)	0.263	10 (35.7%)	14 (50.0%)	0.418
Malignant	21 (67.7%)	14 (50.0%)		18 (64.3%)	14 (50.0%)	
Diameter of resected tumor (mm)						
Mean (SD)	51.6 (33.1)	31.9 (25.7)	0.001	49.7 (32.9)	31.9 (25.7)	0.003
number LNN Retrieved						
Mean (SD)	15.6 (12.8)	24.9 (15.0)	0.013	15.2 (13.5)	24.9 (15.0)	0.011 ^MW^
number Pathologic.LNN						
Mean (SD)	1.34 (2.21)	1.83 (2.71)	0.528	1.50 (2.28)	1.83 (2.71)	0.763 ^MW^
Intraoperative blood transfusion						
No	25 (80.6%)	28 (100%)	0.244	22 (78.6%)	28 (100%)	0.196
Yes	3 (9.7%)	0 (0%)		3 (10.7%)	0 (0%)	
Postoperative ICU stay (days)						
Mean (SD)	2.63 (2.98)	1.64 (2.41)	0.154	2.70 (3.02)	1.64 (2.41)	0.144 ^MW^
POPF &			0.029			0.026
None	21 (67.7%)	26 (92.9%)	ns	19 (67.9%)	26 (92.9%)	ns
Biochemical leak	4 (12.9%)	2 (7.1%)	ns	3 (10.7%)	2 (7.1%)	ns
Grade B/C	6 (19.4%)	0 (0%)	0.08	6 (21.4%)	0 (0%)	0.057
Resection margin						
R0	25 (83.3%)	26 (92.9%)	0.420	22 (78.6%)	26 (92.9%)	0.713 £
R1	5 (16.7%)	2 (7.1%)		5 (17.9%)	2 (7.1%)	
Operative time (min)						
Mean (SD)	273 (80.1)	340 (84.7)	0.011	265 (78.4)	340 (84.7)	0.091 £
Estimated blood loss (mL)						
Mean (SD)	620 (451)	262 (284)	<0.001	603 (471)	262 (284)	0.126 ^MW^ £
Postoperative hospital stay (days)						
Mean (SD)	16.9 (12.4)	11.8 (5.36)	0.057	17.5 (13.1)	11.8 (5.36)	0.001 ^MW^ £
Postoperative Clavien-Dindo morbidity ≥ grade 3						
No	27 (87.1%)	18 (64.3%)	0.062	24 (85.7%)	18 (64.3%)	0.384 £
Yes	4 (12.9%)	10 (35.7%)		4 (14.3%)	10 (35.7%)	
30-day mortality						
No	29 (93.5%)	27 (96.4%)	0.998	27 (96.4%)	27 (96.4%)	0.997 £
Yes	1 (3.2%)	1 (3.6%)		1 (3.6%)	1 (3.6%)	
90-day mortality						
No	16 (51.6%)	27 (96.4%)	0.975 #	13 (46.4%)	27 (96.4%)	0.510 £
Yes	0 (0%)	1 (3.6%)		0 (0%)	1 (3.6%)	

Abbreviation. ODP: open distal pancreatectomy; LDP: laparoscopic distal pancreatectomy; RDP: robotic distal pancreatectomy. ICU = intensive care unit, LNN = lymph node, SD = standard deviation; POPF: postoperative pancreatic fistula. MW: Mann–Whitney test *p*-value; #: chi-squared test on available (excluding missing) data; &: general chi-square *p*-value is reported in the line of the name of the variable. Post-hoc chi-square *p*-values, adjusted for Bonferroni correction, are reported in the lines corresponding to each category of the variable (ns: not significant at level 0.05); £: Since the two groups are statistically different in terms of ‘Diameter of largest tumor’ also after the matching for this variable, the *p*-value was obtained by Generalize linear model adjusting for this variable.

**Table 4 cancers-16-01051-t004:** Laparoscopic vs. Robotic distal pancreatectomy.

	Before Matching			After Matching		
	LDP (*n* = 162)	RDP (*n* = 28)	*p* Value	LDP (*n* = 28)	RDP (*n* = 28)	*p* Value
Gender						
Female	95 (58.6%)	21 (75.0%)	0.153	17 (60.7%)	21 (75.0%)	0.391
Male	67 (41.4%)	7 (25.0%)		11 (39.3%)	7 (25.0%)	
Age (years)						
Mean (SD)	61.7 (14.7)	60.8 (18.1)	0.939	61.3 (14.9)	60.8 (18.1)	0.915
BMI (kg/m^2^)						
Mean (SD)	28.1 (4.18)	25.7 (5.41)	0.004	25.1 (3.40)	25.7 (5.41)	0.961
ASA Score						
1	23 (14.2%)	2 (7.1%)	0.124	3 (10.7%)	2 (7.1%)	0.700
2	102 (63%)	14 (50.0%)	ns	11 (39.3%)	14 (50.0%)	
3	36 (22.2%)	12 (42.9%)	ns	14 (50.0%)	12 (42.9%)	
4	1 (0.6%)		ns			
Prior Abdominal Surgery						
No	132 (81.5%)	19 (67.9%)	0.163	23 (82.1%)	19 (67.9%)	0.355
Yes	30 (18.5%)	9 (32.1%)		5 (17.9%)	9 (32.1%)	
Neoadjuvant Chemotherapy						
No	168 (98.8%)	28 (100%)	0.999	28 (100%)	28 (100%)	1
Yes	2 (1.2%)	0 (0%)		0 (0%)	0 (0%)	
Type of lesion						
Benign	72 (44.4%)	14 (50.0%)	0.734	12(42.9%)	14 (50.0%)	0.789
Malignant	90 (55.6%)	14 (50.0%)		16 (57.1%)	14 (50.0%)	
Diameter of resected tumor (mm)						
Mean (SD)	35.8 (19.9)	31.9 (25.7)	0.128	31.6 (19.1)	31.9 (25.7)	0.902
number LNN Retrieved						
Mean (SD)	10.6 (7.70)	24.9 (15.0)	<0.001	7.69 (5.51)	24.9 (15.0)	<0.001
number Pathologic.LNN						
Mean (SD)	0.94 (1.32)	1.83 (2.71)	0.287	0.55 (0.82)	1.8 (2.71)	0.206
Intraoperative blood transfusion						
No	152 (93.8%)	28 (100%)	0.554	28 (100%)	28 (100%)	1
Yes	7 (4.3%)	0 (0%)		0 (0%)	0 (0%)	
Postoperative ICU stay (days)						
Mean (SD)	0.26 (1.18)	1.64 (2.41)	<0.001	0.63 (1.74)	1.64 (2.41)	<0.001 ^MW^
POPF &			<0.001			<0.001
None	86 (53.1%)	26 (92.9%)	<0.001	14 (50.0%)	26 (92.9%)	<0.001
Biochemical leak	53 (32.7%)	2 (7.1%)	0.024	9 (32.1%)	2 (7.1%)	ns
Grade B/C	23 (14.2%)	0 (0%)	ns	5 (17.9%)	0 (0%)	0.014
Conversion			0.007			0.156
no	152 (89.4%)	18 (64.3%)		24 (85.7%)	18 (64.3%)	
yes	7 (4.1%)	5 (17.9%)		1 (3.6%)	5 (17.9%)	
Resection margin						
R0	141 (87%)	26 (92.9%)	0.577	27 (96.4%)	26 (92.9%)	0.993
R1	21 (13%)	2 (7.1%)		1 (3.6%)	2 (7.1%)	
Operative time (min)						
Mean (SD)	210 (73.1)	340 (84.7)	<0.001	210 (66.7)	340 (84.7)	<0.001
Estimated blood loss (mL)						
Mean (SD)	282 (355)	262 (284)	0.884	222 (215)	262 (284)	0.817 ^MW^
Postoperative hospital stay (days)						
Mean (SD)	6.38 (5.01)	11.8 (5.36)	<0.001	6.61 (4.17)	11.8 (5.36)	<0.001 ^MW^
Postoperative Clavien-Dindo morbidity ≥ grade 3						
No	139 (85.8%)	18 (64.3%)	0.010	21 (75.0%)	18 (64.3%)	0.561
Yes	23 (14.2%)	10 (35.7%)		7 (25.0%)	10 (35.7%)	
30-day mortality						
No	160 (98.8%)	27 (96.4%)	0.924	27 (96.4%)	27 (96.4%)	1
Yes	2 (1.2%)	1 (3.6%)		1 (3.6%)	1 (3.6%)	
90-day mortality						
No	160 (98.8%)	27 (96.4%)	0.924	27 (96.4%)	27 (96.4%)	1
Yes	2 (1.2%)	1 (3.6%)		1 (3.6%)	1 (3.6%)	

Abbreviation. ODP: open distal pancreatectomy; LDP: laparoscopic distal pancreatectomy; RDP: robotic distal pancreatectomy. ICU = intensive care unit, LNN = lymph node, SD = standard deviation; POPF: postoperative pancreatic fistula. MW: Mann–Whitney test *p*-value; &: general chi-square *p*-value is reported in the line of the name of the variable. Post-hoc chi-square *p*-values, adjusted for Bonferroni correction, are reported in the lines corresponding to each category of the variable (ns: not significant at level 0.05).

## Data Availability

Data available upon reasonable request.

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
