# Peer review of "Study International Multicentric Pancreatic Left Resections (SIMPLR): Does Surgical Approach Matter?"

_cancers, 2024, doi:10.3390/cancers16051051_

Round 1

Reviewer 1 Report

Comments and Suggestions for Authors

I thank the editors for the opportunity to review the manuscript by Acciuffi et al. The authors describe a retrospective analysis of postoperative outcomes after open vs. laparoscopic vs. distal pancreatectomy in three HPB centers in Europe. All in all, they present an interesting, easy-to-read and important manuscript. A publication in Cancers might be considered, however, several flaws need to be addressed:

Major flaws:

1. The authors should describe the surgical procedures in more detail. Were there differences between the centers? Have they tried to standardize the procedures between the centers?

2. Why was the BMI not used as a parameter for PSM? BMI is a known predictor of a higher difficulty for robotic surgery. In fact, patients in the open group had a higher mean BMI in the un-matched comparison. After matching, BMI is still quite different between open and laparoscopic, and significantly different between laparoscopic and robotic. In addition, the authors failed to report the patients' comorbidities, which should also be part of the matching process. Please explain and report pre-existing conditions (hypertension, pulmonary disease, etc). Having failed to implement BMI and comorbidities heavily impacts the quality of PSM, and therefore the results of this analysis.

3. All tables need major revisions. Units are missing. Use of acronyms is recommended (ODP etc.). Some variables have dots instead of spaces (e. g. "type.of.lesions"). Some symbols are not declared (#, $). In some cases, p values are missing, instead "ns" is reported, which is also not declared. I recommend thorough proof-reading by all authors.

4. The numbers of patients between the matched and un-matched analysis are not identical (e. g. ODP: 34 un-matched vs. 31 matched; the same applies for LDP and RDP). The authors need to explain and specify, why patients were excluded from PSM.

Minor flaws:

1. In many parts of the manuscript, the units of reported measures are missing (abstract, tables, results). I highly recommend thorough proof-reading by all authors.

2. Please check the manuscript for not described abbreviations (e. g. l. 120 "LOS").

3. Statistical analysis: Means are presented for continuous data, however, I doubt that all distributions are parametric. I recommend using medians.

4. I recommend using the acronyms (ODP, LDP, RDP) consistently throughout the manuscript.

Comments on the Quality of English Language

Overall, English language is fine. Minor adjustments need to be made, for example l. 59 (poor language). I recommend proof reading by a native speaker.

Reviewer 2 Report

Comments and Suggestions for Authors

The study from Sara Acciuffi et al. examines the outcomes of different surgical techniques for left pancreatic resection, comparing open, laparoscopic, and robotic approaches in terms of perioperative and oncological outcomes. It utilizes propensity score matching analysis to assess these outcomes in patients from high-volume centers. The research aims to determine the most effective surgical method, with a focus on minimizing complications and improving patient prognosis. The findings are expected to offer insights into the best practices for pancreatic surgery, potentially influencing future surgical approaches in this field. While this study provides innovative insights into the role of different surgical techniques in pancreatic resection, several issues need to be addressed to enhance the quality of this research.

1.     Emphasize the clinical implications of the findings, especially in the choice of surgical approach.

2.     Discuss the potential limitations and biases, including the retrospective nature and selection bias.

3.     The manuscript lacks a clear explanation of the sample size determination. Including a power analysis would ensure that the study is adequately powered to detect significant differences between surgical methods.

4. There should be a clearer explanation of how missing data was handled. If imputation methods were used, the technique and rationale should be described. 

Comments on the Quality of English Language

 Minor editing of English language required
